# Fuse Your Latents: Video Editing with Multi-source Latent Diffusion Models

Submission Id: 1490

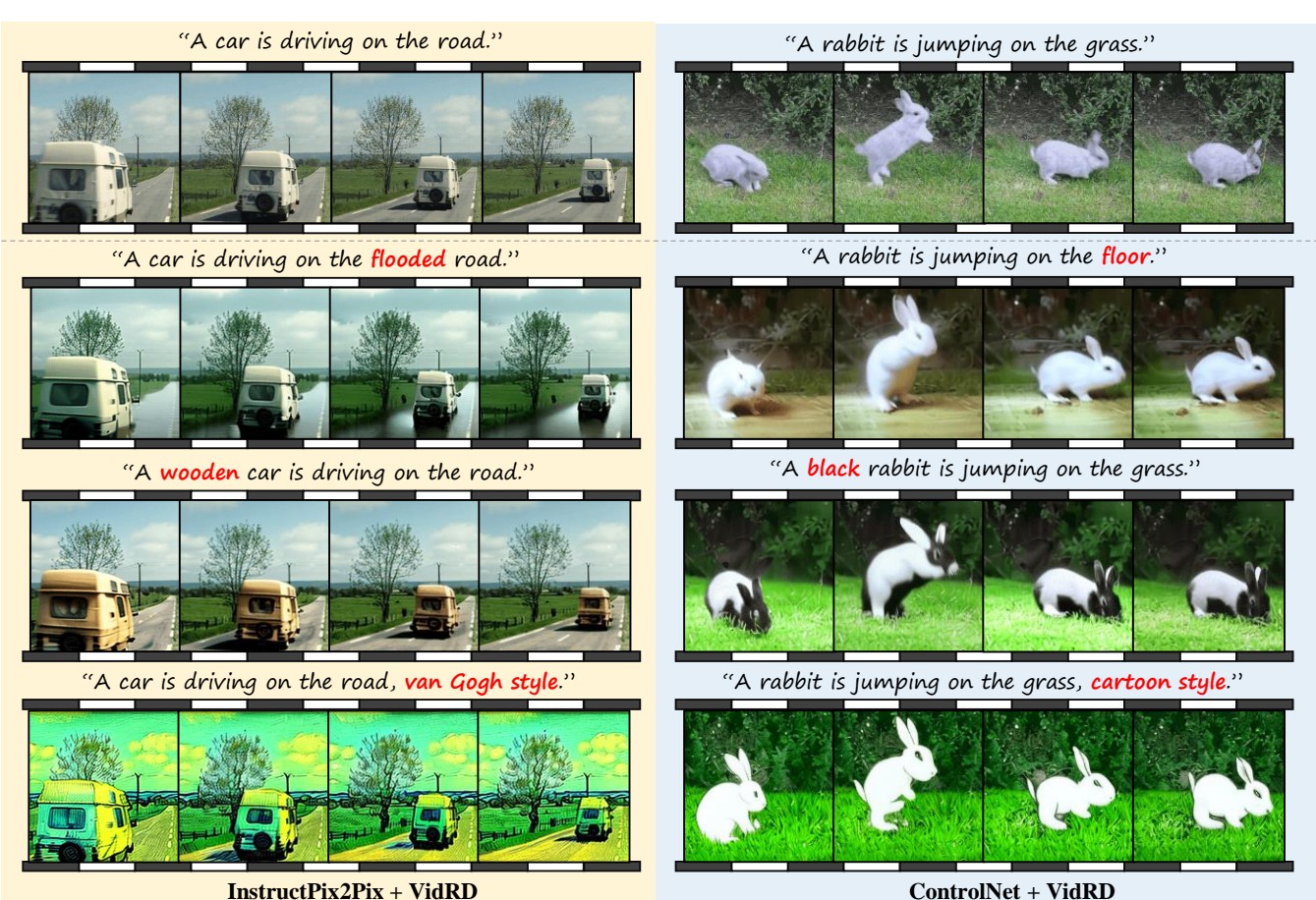

**Figure 1: FLDM can serve as a versatile plugin that can be applied to off-the-shelf image diffusion models (e.g., InstructPix2Pix [3] and ControlNet [50]) and video diffusion models (e.g., VidRD [12]).**

## ABSTRACT

Latent Diffusion Models (LDMs) are renowned for their powerful capabilities in image and video synthesis. Yet, compared to text-to-image (T2I) editing, text-to-video (T2V) editing suffers from a lack of decent temporal consistency and structure, due to insufficient pre-training data, limited model editability, or extensive tuning costs. To address this gap, we propose FLDM (Fused Latent Diffusion Model), a training-free framework that achieves high-quality T2V editing by integrating various T2I and T2V LDMs. Specifically, FLDM utilizes a hyper-parameter with an update schedule to effectively fuse image and video latents during the denoising process. This paper is the first to reveal that T2I and T2V LDMs can complement each other in terms of structure and temporal consistency, ultimately generating high-quality videos. It is worth noting that FLDM can serve as a versatile plugin, applicable to off-the-shelf image and video LDMs, to significantly enhance the quality of video editing. Extensive quantitative and qualitative experiments on popular T2I and T2V

LDMs demonstrate FLDM's superior editing quality than state-of-the-art T2V editing methods. Our project page is available at https://anonymous121381.github.io/FLDM/.

## CCS CONCEPTS

• **Computing methodologies** → **Computer vision**.

## KEYWORDS

Video editing, latent diffusion models, training-free framework

## 1 INTRODUCTION

Recently, diffusion models [16, 42] have achieved significant success in image generation [7, 10, 20, 22, 29, 37, 38, 40], image-to-image translation [39], text-to-image editing [3, 14, 18, 29] and image inpainting [21, 25, 36]. The development of text-to-image (T2I) generation and editing models has inspired text-to-video (T2V) editing. A straightforward way of T2V editing is to utilize T2I models, considering temporal consistency modeling with motion conditions [4, 6], temporal propagation [5, 11, 17, 30] and temporal modules [13, 32, 46]. Although T2I models have the advantage of video editing with high-quality structure and appearance, the direct adaptation from T2I models to T2V editing still has limitations in temporal consistency if the edited target has significant changes in shape [32] or motion [30]. Besides, the one-shot tuning strategy [1, 28, 46] suffers from low editability and training efficiency.

Therefore, another option is to develop video generation diffusion models [2, 9, 15, 44] for video editing that can achieve decent temporal consistency. Compared to the images generated by T2I models, the videos generated by T2V generation models, despite having good temporal consistency, still lack structural integrity (e.g., textures, low resolution). The main reason is that obtaining high-quality video data is more challenging compared to image data, resulting in a smaller amount of video data available for training T2V generation models [44, 45]. Additionally, to achieve high-quality video generation, T2V models require the incorporation of numerous conditional controls during training, such as depth and optical flow [9, 44], which increases the training cost compared to T2I generation models.

To this end, to achieve high-quality video editing, typically evaluated with temporal consistency, text alignment, and aesthetics, the question arises: Given T2I and T2V diffusion models, how can we maximize their potential for superior T2V editing with minimal operations? In this paper, for the first time, we investigate how to ensemble T2I and T2V diffusion models for video editing. We introduce the FLDM (Fused Latent Diffusion Model) by fusing multi-source LDMs, including T2I and T2V latent diffusion models. Drawing inspiration from Prompt2Prompt [14], certain video editing methods such as FateZero [32] facilitate local editing through attention map manipulation, which can achieve fine-grained editing results. However, this way cannot guarantee robust temporal coherence and suffers from low editability. FLDM replaces attention injection with latent fusion, which realizes an overall and global editing.

Concretely, we extract latents from both T2V and T2I LDMs. At every denoising timestep, we apply latent fusion with a hyper-parameter to control the proportion of image to video latents. The

underlying intuition is that this allows us to balance temporal consistency against structural integrity. The fused latents, enriched with both temporal and semantic information from the respective models, will undergo individual denoising by each model in the subsequent timestep. This suggests that both models can leverage supplemental information—namely, temporal modeling from T2V LDMs and superior generation quality from T2I LDMs. However, the excessive guidance from T2I LDMs may introduce image artifacts during the latent fusion process. To mitigate such side effects, we adjust the fusion ratio for FLDMs during the denoising process with an update schedule. Through extensive experiments with a range of T2I and T2V LDMs, we've found that while T2V LDMs struggle to maintain the structural integrity of the source video, they are greatly enhanced by the complementary strengths of T2I LDMs. Conversely, T2V LDMs excel in ensuring robust temporal consistency in the edited videos, thereby elevating the overall quality of edits performed by T2I LDMs. Figure 1 displays several video editing results using FLDM with InstructPix2Pix [3] and ControlNet [50].

To sum up, we make the following contributions.

- We propose multi-source latent fusion, a straightforward yet effective inference strategy that can seamlessly integrate off-the-shelf T2I diffusion models with T2V diffusion models.
- To the best of our knowledge, this work is the first to reveal that T2I and T2V diffusion models can complement each other effectively. Specifically, T2I models are essential for providing structural integrity, while T2V models are crucial for ensuring temporal consistency.
- Without any tuning process, our method achieves better performance than other counterparts on both qualitative and quantitative evaluations.

## 2 RELATED WORK

### 2.1 Text-to-Image Generation

Before the advent of diffusion models, prominent text-to-image (T2I) and text-to-video (T2V) generation works predominantly adopted GAN [35, 49] or auto-regressive architectures [8, 34, 48]. Dhariwal et al. [7] were among the first to present comparisons between GANs and Diffusion Models in the context of image synthesis. Their findings suggested that diffusion models outperformed GANs in terms of diversity, stability of training objectives, and scalability. Subsequently, a series of text-to-image generation models based on the diffusion approach emerged [3, 18, 27, 29, 37, 38, 40, 50], achieving high-fidelity generation.

While T2I diffusion models have achieved remarkable performance in T2I generation, directly adapting these models for T2V editing proves insufficient due to the absence of temporal consistency modeling. Our method combines latents from both T2I and T2V models to enhance temporal consistency without compromising the editing capability of the T2I model. With no additional training or tuning required, the introduced latent fusion strategy is adaptable to any off-the-shelf T2I or T2V diffusion model, enhancing editing quality with minimal adjustments.

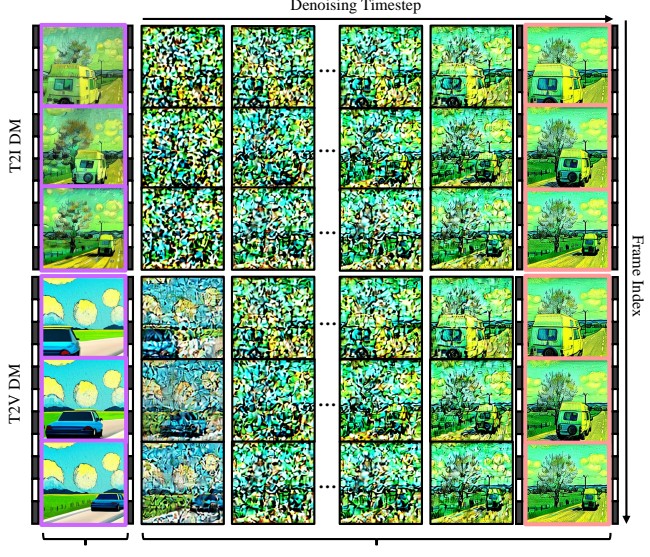

**Figure 2: Denoising process of T2I and T2V LDMs (+ Van Gogh style). First column: Without FLDM, T2I LDMs have good structure preservation but lack temporal consistency, T2V LDMs lack structure preservation but achieve good temporal consistency. Last column: Both the structure and temporal consistency of the edited videos are enhanced with multi-source latent fusion (FLDM). Best viewed from project homepage.**

## 2.2 Diffusion Models for Video Editing

A prevalent approach to video editing using diffusion models involves adapting T2I models to T2V models [41, 46]. This adaptation often incorporates temporal modules to guarantee temporal consistency. Tune-A-Video [46] integrates temporal attention layers into UNet and conducts one-shot tuning. Meanwhile, Make-A-Video [41] augments the network to encompass temporal information by extending it with spatial-temporal modules. However, fine-tuning target videos can lead to over-fitting source prompts, potentially diminishing the model's editing capabilities. To address this, several studies employ T2V diffusion models for video editing and demonstrate promising results. [28] introduces a mixed fine-tuning strategy for the Imagen Video model [15] that enhances motion editing. [9] presents a video diffusion model trained with depth information to govern video structure and content. Although these methods have yielded remarkable results in video editing, training T2V diffusion models poses a challenge due to the inherent difficulty in collecting high-quality video data for training. Furthermore, their primary focus is on enhancing T2V editing through mixed video-image fine-tuning and joint video-image training, neglecting the potential benefits of leveraging existing high-quality T2I models.

Another avenue of research is inspired by Prompt2Prompt [14] and Plug-and-Play [43], both of which facilitate local editing through attention map manipulation. [32] suggests blending self-attention maps with masks produced by cross-attention maps to support zero-shot video editing. Meanwhile, [24] introduces a decoupled-guidance attention control, adapting P2P to Video-P2P. However, these methods require extensive manual adjustment of the parameters in the attention map, making them challenging to apply in practical scenarios.

In this paper, we introduce a simple-yet-effective latent fusion strategy that harnesses the strengths of both T2I and T2V diffusion models for T2V editing without requiring any tuning. For the first time, this study demonstrates that T2I and T2V diffusion models complement each other.

## 3 METHOD

Our objective is to maximize the advantages of the existing text-to-image (T2I) and text-to-video (T2V) diffusion models in terms of structure integrity and temporal consistency, aiming to achieve high-quality T2V editing. Given a video $\mathcal{V}$ containing $m$ frames, denoted as $\mathcal{V} = \{\mathbf{x}_i \mid i \in [1, m]\}$, T2V editing aims to generate a new video conditioned on an editing text prompt $P$. In this section, we first introduce the preliminaries of the diffusion model in Section 3.1, including latent diffusion models [37] and DDIM inversion [42]. Then the detail design of FLDM (Fused Latent Diffusion Model) will be introduced in Section 3.2 and Section 3.3.

### 3.1 Preliminaries

**Latent Diffusion Models (LDMs).** LDMs denoise noisy latents in the latent space. First, an image $\mathbf{x}$ undergoes compression via an encoder $\mathcal{E}$, producing a latent representation $\mathbf{z} = \mathcal{E}(\mathbf{x})$, which is reconstructed back into an image through a decoder $\mathcal{D}$. A U-Net $U_\theta$ is applied as the denoising model to predict the noise, which is optimized by minimizing the following objective

$$\min_{\theta} E_{\boldsymbol{\epsilon} \sim \mathcal{N}(\mathbf{0}, \mathbf{I}), t \sim \mathcal{U}(1, T)} \| \boldsymbol{\epsilon} - U_\theta(\mathbf{z}_t, t, P) \|_2^2, \quad (1)$$

where $\mathbf{z}_t$ is the noisy latent at timestep $t$ and $P$ is the conditional text embedding.

**DDIM Inversion.** Deterministic DDIM sampling inverts the noisy latent $\mathbf{z}_T$ to a clean latent $\mathbf{z}_0$, the sampling process is expressed as

$$\mathbf{z}_{t-1} = \sqrt{\bar{\alpha}_{t-1}} \frac{\mathbf{z}_t - \sqrt{1 - \bar{\alpha}_t} \hat{\boldsymbol{\epsilon}}}{\sqrt{\bar{\alpha}_t}} + \sqrt{1 - \bar{\alpha}_{t-1}} \hat{\boldsymbol{\epsilon}} \quad (2)$$

---

**Algorithm 1** FLDM for T2V Editing

**Input**: Latent features from DDIM inversion by T2V and T2I model: $\mathbf{z}_T^V$, $\mathbf{z}_T^I$, target prompts $P$, fusion timestep $\tau$, fusion ratio $\alpha_\tau$.

**Output**: Denoised latent $\mathbf{z}_0^*$.

1: **for** $t = T, T - 1, ..., 1$ **do**
2:      $\mathbf{z}_{t-1}^I \leftarrow \text{DDIM}_{\text{T2I}}(\mathbf{z}_t^I, P, t);$
3:      $\mathbf{z}_{t-1}^V \leftarrow \text{DDIM}_{\text{T2V}}(\mathbf{z}_t^V, P, t);$
4:      **if** $t \leq (T - \tau)$ **then**
5:          $\mathbf{z}_{t-1}^* = \alpha_t * \mathbf{z}_{t-1}^V + (1 - \alpha_t) * \mathbf{z}_{t-1}^I;$
6:          $\alpha_{t-1} = \alpha_t + (1 - \alpha_\tau)/(T - \tau);$
7:          $\mathbf{z}_{t-1}^I \leftarrow \mathbf{z}_{t-1}^*;$
8:          $\mathbf{z}_{t-1}^V \leftarrow \mathbf{z}_{t-1}^*;$
9:      **end if**
10: **end for**
11: **return** $\mathbf{z}_0^*$

---

**Figure 3: FLDM framework for T2V editing. During the inference stage, the input video is encoded via VAE Encoder to be a clean latent $z_0 \in \mathbb{R}^{f \times c \times h \times w}$ and then inverted to be a noisy latent $z_T \in \mathbb{R}^{f \times c \times h \times w}$ through DDIM inversion. During the first $\tau$ timesteps, the T2V LDM and T2I LDM predict noise for noisy latent respectively. In the next $T - \tau$ timesteps, a multi-source latent fusion module is applied to fuse denoised latents from T2V and T2I LDMs.**

where $\hat{\epsilon}$ is the predicted noise and $\alpha_t$ is the multiplication of variances at timestep $t$. DDIM inversion is the reverse process of DDIM sampling, it converts a clean latent $\mathbf{z}_0$ to a noised latent $\hat{\mathbf{z}}_T$ which can preserve the structure information in real images.

$$\hat{\mathbf{z}}_t = \sqrt{\bar{\alpha}_t} \frac{\hat{\mathbf{z}}_{t-1} - \sqrt{1 - \bar{\alpha}_{t-1}}\hat{\epsilon}}{\sqrt{\bar{\alpha}_{t-1}}} + \sqrt{1 - \bar{\alpha}_t}\hat{\epsilon} \quad (3)$$

## 3.2 Multi-source Latent Fusion

As depicted in Figure 2, video editing conducted separately by T2I and T2V LDMs, without the integration of FLDM, reveals distinct drawbacks. T2I LDMs, although adept in frame-wise editing, tend to cause flickering issues, e.g., the tree in the background has obvious shape changes across frames. Similarly, T2V LDMs, despite their ability to maintain temporal consistency, often compromise on structural quality, e.g., the car shape is different from the original car (see the first row in Figure 1). FLDM addresses these issues by strategically fusing the latents of both T2I and T2V LDMs during the denoising process. As the denoising timesteps progress, there is an improvement in both the structural integrity and temporal consistency of the video edits.

Figure 3 shows the FLDM pipeline of video editing with multi-source latent fusion. Given the source video $\mathcal{V}_{\text{src}}$ and a target prompt $P$, the VAE Encoder compresses the input video into a clean latent $\mathbf{z}_0 \in \mathbb{R}^{f \times c \times h \times w}$, where $f$ represents the number of frames in the source video. Then we apply DDIM inversion with T2V and T2I LDMs, which makes use of the source prompt, to invert the clean latent into noisy latents $\mathbf{z}_T^V \in \mathbb{R}^{f \times c \times h \times w}$ and $\mathbf{z}_T^I \in \mathbb{R}^{f \times c \times h \times w}$. Afterward, T2V and T2I LDMs are applied to predict noise. Note that the T2I LDMs predict noise frame-by-frame and the denoised T2I latents are concatenated at the temporal dimension. Then we apply latent fusion by combining the denoised latents from T2V and T2I LDMs. As a result, a mixed noisy latent $\mathbf{z}_t^*$ is generated for the next denoising timestep

$$\mathbf{z}_t^* = \alpha_t \mathbf{z}_t^V + (1 - \alpha_t)\mathbf{z}_t^I \quad (4)$$

where the hyper-parameter $\alpha$ controls the ratio of T2V and T2I latents. A larger $\alpha$ ensures better temporal consistency but does

harm to structure and text alignment, while if $\alpha$ is too small, it would degrade to frame-wise editing with T2I LDMs.

Notice that, our method is a versatile plugin that can work for latent diffusion models within the same latent space. We need to ensure that two LDMs share the same VAE Encoder and Decoder. Most LDMs are trained on the basis of Stable Diffusion [37], where the weights of VAE are frozen and remain unchanged. As we utilize off-the-shelf T2I LDMs and T2V LDMs for joint denoising, the editing results depend on the fused models. Specifically, the temporal modeling capability of T2V LDMs affects the temporal consistency of edited video frames, while the spatial structure of the videos is related to T2I LDMs.

## 3.3 Fusion Ratio Update Schedule

In practice, we notice that fusing multiple latents at the early diffusion steps results in noisy output, since latent fusion may break the structure of videos. As a result, we let T2V and T2I LDMs infer separately in early $\tau$ steps and perform latents fusion at afterward steps. Although $\alpha$ balances the ratio of temporal consistency and structure modeling, a fixed $\alpha$ may bring too much guidance from T2I LDMs at the end of the denoising process, thus causing temporal inconsistency in generated videos. Since T2V LDMs are able to generate temporal coherent videos, we increase the ratio of T2V latents at the end of the denoising process. We propose an update schedule for the fusion ratio, wherein the ratio is incrementally increased after each latent fusion step as following:

$$\alpha_{t-1} = \alpha_t + (1 - \alpha_\tau)/(T - \tau) \quad (5)$$

where $T$ is the total number of denoising steps, $\alpha_t$ is the fusion ratio at timestep $t$. In practice, we start to increase $\alpha_t$ at timestep $\tau$. The initial fusion ratio $\alpha_\tau$ ranges between $[0, 1]$. Formally, the algorithm is shown in Algorithm 1.

FLDM aims to augment the capability of T2V models through T2I models, ensuring that T2V models retain a prominent position. So T2V latents should be allocated a larger propotion. The T2V

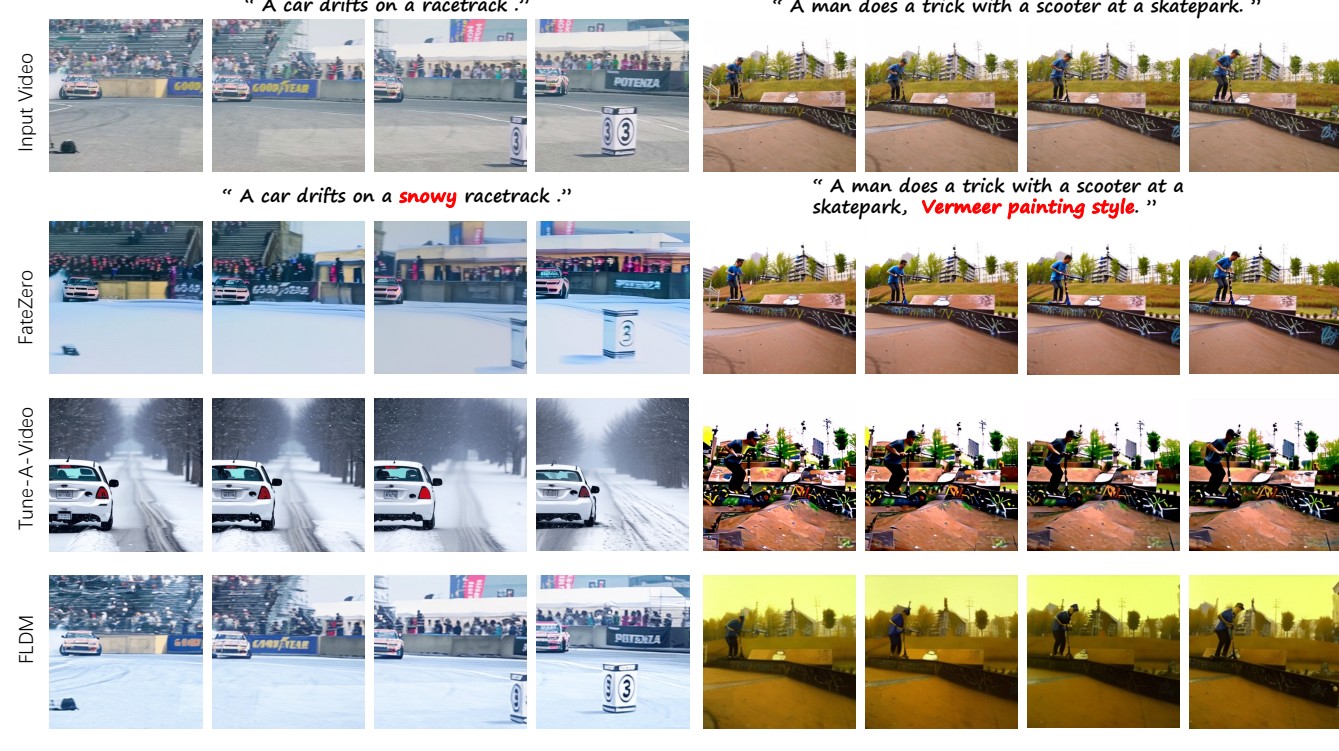

**Figure 4: Qualitative comparison with SOTA approaches. FLDM has the best textual alignment, temporal consistency, and fidelity.**

signals and T2I signals in fused latents can be calculated as:

$$S_V = \int_\tau^T \alpha_t dt \qquad (6)$$

$$S_I = \int_\tau^T (1 - \alpha_t) dt \qquad (7)$$

As $\alpha_t$ changes in a linear manner, we can calculate the difference between the T2V and T2I contributions:

$$\delta = S_V - S_I = 2\alpha_\tau(T - \tau) > 0 \qquad (8)$$

which means T2V signals take a dominant role. On the other hand, a linear update schedule ensures a smoother denoising process than a polynomial way.

## 4 EXPERIMENTS

### 4.1 Experiment Settings

**Implementation Details.** We attempt to apply FLDM to various T2I and T2V diffusion models to validate the effectiveness. For the diffusion model gallery, we consider popular T2I diffusion models including ControlNet [50] and InstructPix2Pix [3] loaded with pre-trained weights from Stable Diffusion v1.5 [37]. Although there are some high-quality video diffusion models [9, 15, 45], unfortunately, they can not be accessed. Therefore, we take the publicly available VidRD [12] and ZeroScope [26] with released pre-trained weights. We then sample 8 frames uniformly from input videos with a resolution of 256p for all models. For T2V models fused

with ControlNet, we use ControlNet with canny edge condition and set the total timestep $T = 50$ for DDIM schedule. For T2V models fused with InstructPix2Pix, we set the total timestep $T = 100$ and the classifier-free guidance scale of text to 12.5 and of image to 1.5. We select 16 videos from DAVIS dataset [31] for evaluation, which are part of the TGVE benchmark [47]. For each video, there are three types of text prompts for video editing including *style*, *object*, and *background*.

**Evaluation Metrics.** Following previous text-to-video editing works [9, 32, 46], we conduct both quantitative and qualitative evaluations for FLDM. For quantitative evaluation, we utilize automatic metrics including frame consistency (**'Tem-Con'**), text alignment (**'Text-Align'**), user preference (**'User-Pre'**) with CLIPScore [33] and PickScore [19]. For user study, three metrics (denoted as **'Edit'**, **'Image'**, and **'Temp'**) are conducted to evaluate editing quality, frame fidelity, and temporal consistency of edited video. Following [23], we recruit 20 evaluators to pair-wisely compare our method with baselines, and present the preference percentage with "$p_1/p_2$" where $p_1$ denotes the preference percentage of baseline, $p_2$ denotes the preference percentage of our method.

### 4.2 Main Results

**Baselines.** We test two T2I models ControlNet (CN) and Instruct-Pix2Pix (IP2P), and two T2V models VidRD and ZeroScope (ZS) under the individual testing setting (T2I or T2V) and FLDMs testing setting (T2I + T2V). In addition, we compare our method against two

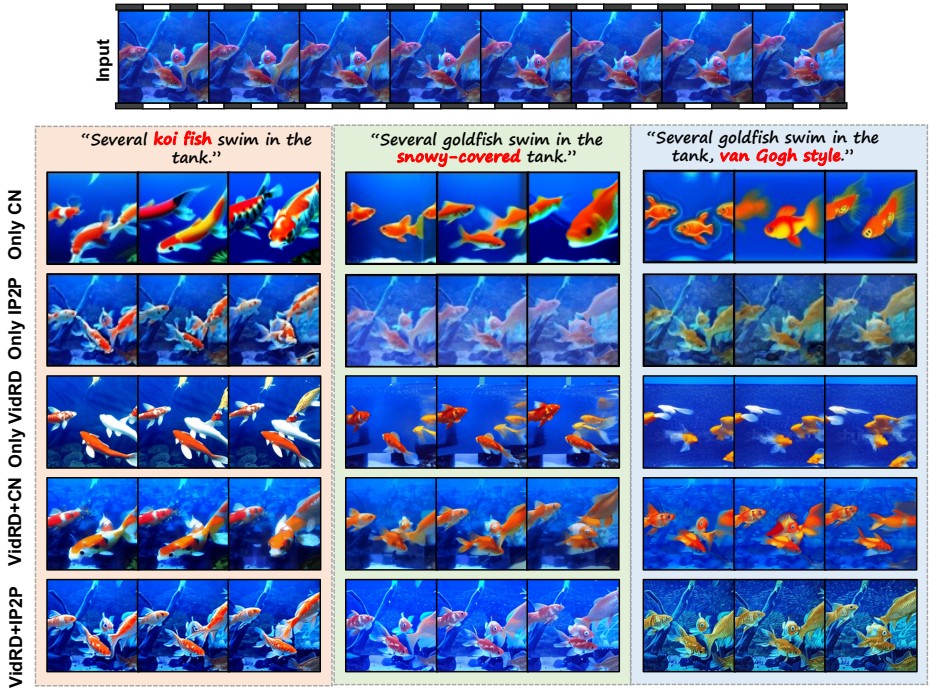

**Figure 5: Object, background, and style editing results of FLDM. In comparison with T2I models only, FLDM with VidRD generates more consistent frames. Compared to videos edited by VidRD only, FLDM is superior in structure and fidelity.**

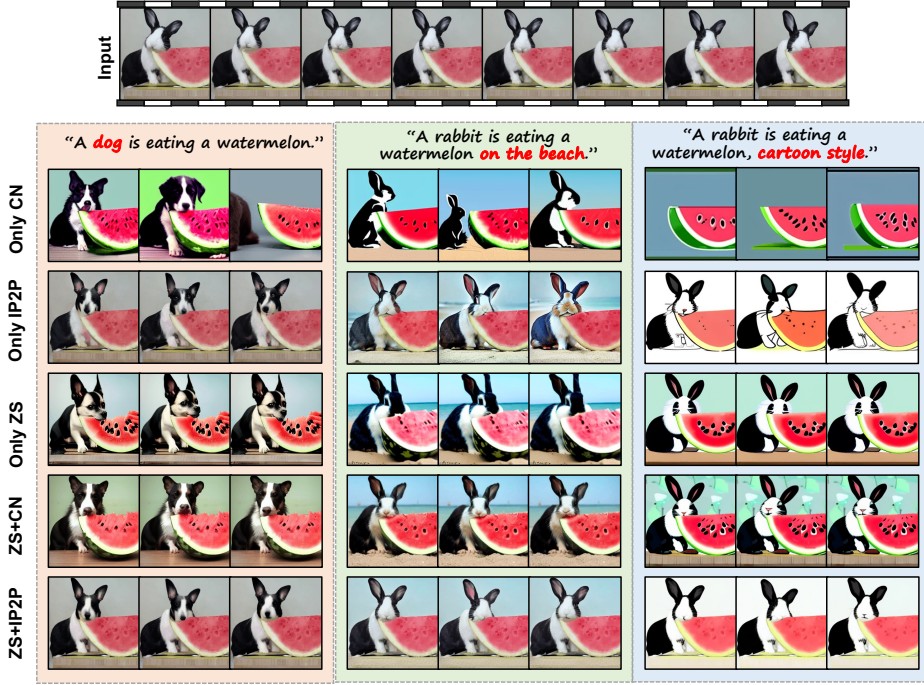

**Figure 6: Results of object, background, and style editing using FLDM with ZeroScope. FLDM is effective for improving editing fidelity and is flexible for various T2I and T2V diffusion models.**

**Table 1: Automatic evaluation of FLDMs and other baseline methods. FLDMs achieve the best textual alignment, user preference, and comparable temporal consistency.**

| Method | CLIP Mertics | | PickScore |
|---|---|---|---|
| Inversion & Editing | Tem-Con | Text-Align | User-Pre |
| Framewise IP2P [3] | 86.76 | 25.11 | 20.33 |
| Framewise CN & DDIM [50] | 80.02 | 25.41 | 20.12 |
| Tune-A-Video & DDIM [46] | 90.45 | 27.13 | 20.42 |
| FateZero [32] | **92.92** | 23.81 | 20.22 |
| VidRD & DDIM [12] | 92.03 | 27.18 | 20.58 |
| ZS & DDIM [26] | 90.24 | 27.49 | 20.58 |
| *FLDMs* | | | |
| VidRD + IP2P | 89.28 | 26.71 | 20.66 |
| VidRD + CN | 90.14 | 27.43 | 20.64 |
| ZS + IP2P | 90.12 | 27.50 | **20.68** |
| ZS + CN | 88.24 | **27.70** | 20.60 |

**Table 2: User preference of FLDMs and other baseline methods. FLDMs achieve the highest human preference over all evaluation metrics and outperform all baselines by a clear margin.**

| Method | User Study | | |
|---|---|---|---|
| Inversion & Editing | Edit | Image | Temp |
| Framewise IP2P [3] | 31.60 / **68.40** | 47.90 / **52.10** | 5.25 / **94.75** |
| Framewise CN & DDIM [50] | 21.10 / **78.90** | 10.50/ **89.50** | 10.50 / **89.50** |
| Tune-A-Video & DDIM [46] | 33.32 / **66.68** | 20.18 / **79.82** | 18.43 / **81.57** |
| FateZero [32] | 35.10 / **64.90** | 39.63 / **60.37** | 35.10 / **64.90** |
| VidRD & DDIM [12] | 32.65 / **67.35** | 5.25 / **94.75** | 35.80 / **64.20** |
| ZS & DDIM | 34.75 / **65.25** | 39.50 / **60.50** | 40.25 / **59.75** |

state-of-the-art video editing methods including Tune-A-Video [46] and FateZero [32].

**Applications.** We show three example applications including 1) *Object editing*: Our method has a significant use case in altering objects by manipulating text prompts. This enables effortless replacement, addition, or removal of objects. For example, we can replace "goldfish" with "koi fish" or "sharks" (see Figure 5). When using our method with InstructPix2Pix, the instruction can be "replace the goldfish with koi fish" or just "turn them sharks". 2) *Background change*: Another application of our method is to change the background of video. When altering the location of the object, our method maintains the coherence of the object's motion. For example, we can modify the background of the rabbit in Figure 1 to be "on the floor", and add shadow of the rabbit that doesn't exist in the origin video. Figure 5 shows the results of editing "tank" to be "snowy-covered tank". 3) *Style transfer*: Through taking the knowledge from T2I models in open domains, FLDM can facilitate transforming videos into diverse styles that are challenging to acquire solely from video data. For example, we transform real-world videos into cartoon style (Figure 1), or Van Gogh style (Figure 5), by adding style descriptors to the prompt.

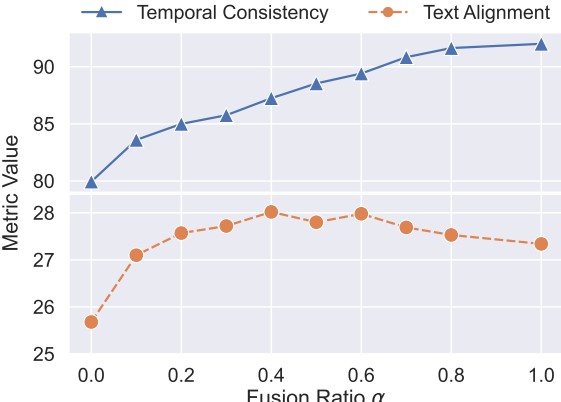

**Figure 7: Temporal consistency and text alignment result with various fusion ratios $\alpha$. Temporal consistency can be improved with larger $\alpha$, and T2V and T2I models complement each other in textual alignment.**

**Qualitative Results.** We showcase visual comparisons of our method against other baselines from Figure 4 to Figure 6. In Figure 5 and Figure 6, we present the effects of fusing T2I models with VidRD and ZeroScope separately. In the case of goldfish, InstructPix2Pix has superior capabilities of frame-wise editing, while lacking consistency between frames (e.g., the appearance of koi fishes exhibits significant variations). In contrast, videos generated by VidRD individually have decent temporal consistency but lack original structure (e.g., background details). When applying FLDM to VidRD and InstructPix2Pix, both temporal consistency and structural integrity can be improved. In Figure 6, we observed that ZeroScope (ZS) achieves satisfactory video editing results with DDIM inversion. However, FLDM can further enhance the visual aesthetics and fidelity of the generated videos (e.g., the cartoon rabbit generated by FLDM is more similar to the original rabbit). In Figure 4, we compare FLDMs with two state-of-art video editing methods. We find Tune-A-Video struggles to keep the fidelity of the original video (e.g., the car is different from the one in the original video) since it lacks regional control. Besides, videos generated by FateZero fail to align with target prompts well (e.g., Vermeer painting style) without carefully tuning on word strength hyper-parameters. It is worth noting that FLDM is more efficient than SOTAs since it does not require any model tuning [46] or heavy handcrafted hyper-parameter tuning [32].

**Quantitative Results.** We use automatic metrics to quantify our method against baselines in Table 1. Results indicate that compared to editing with T2V models only, FLDM achieves higher user preference and better textual alignment at the cost of a slight loss of temporal consistency. FateZero demonstrates good frame consistency but struggles to achieve accurate text-guided editing. Tune-A-video excels in both temporal consistency and textual alignment. In comparison, our method outperforms these two methods in textual alignment and achieves comparable temporal consistency. The user study results are shown in Table 2, which indicates our method is preferred by users across all metrics.

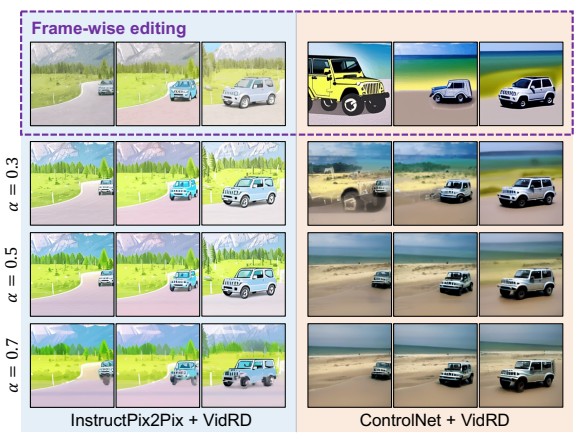

Figure 8: Ablation of Initial Fusion Ratio $\alpha$. The editing prompts are "*a jeep car is moving on the road, cartoon style.*" and "*a jeep car is moving on the beach.*".

## 4.3 Ablation Study

For all the ablation experiments, we take VidRD as the T2V model, which is fused with T2I models (ControlNet and InstructPix2Pix) through FLDM if not other mentioned.

**Ablation of Initial Fusion Ratio $\alpha$.** We conduct extensive experiments with various fusion ratios $\alpha$ as shown in Figure 7 and have the following observations: 1) For temporal consistency, the larger $\alpha$ makes T2V latents have larger weight in fused latents which advances temporal consistency for generated videos. 2) For text alignment, there is one optimal value between [0, 1] which indicates that T2V and T2I models are complementary for text alignment, probably because the T2V model is trained with large-scale video datasets that have rich action concepts in complementary with T2I text prompts [45]. Figure 8 shows some samples generated with two T2I architectures fused with the T2V model. As we can see, the frame-wise editing (only T2I) performs worst since it ignores the temporal relation among frames. While with a T2V model and a suitable fusion ratio $\alpha$, the temporal consistency can be improved clearly. Since training a T2V diffusion model is expensive in collecting datasets and computational cost, we can make the most use of existing T2I models to enhance T2V editing with FLDM. Considering the trade-off between temporal consistency and text alignment, we fuse a smaller proportion of T2I latents compared to T2V latents in practice.

**Ablation of $\tau$.** Table 3 shows the quantitative results of different $\tau$. Figure 9 illustrates the effectiveness of hyper-parameter $\tau$ which decides from which denoising timestep we start to perform FLDM. When FLDM starts at an early timestep (e.g., $\tau <= 30$ for InstructPix2Pix), it may destroy the edited video structure since T2V latents are incorporated into the denoising process too early. If FLDM starts too late (e.g., $\tau >= 45$ for ControlNet), the T2I latents would take too much proportion throughout the denoising process, as a result, destroy temporal consistency for the edited video. As shown in Figure 9, the car in the bottom row exhibits artifacts.

**Ablation of $\alpha$ Update Schedule** As we can see from Figure 10, the $\alpha$ update schedule can improve temporal consistency, e.g., the

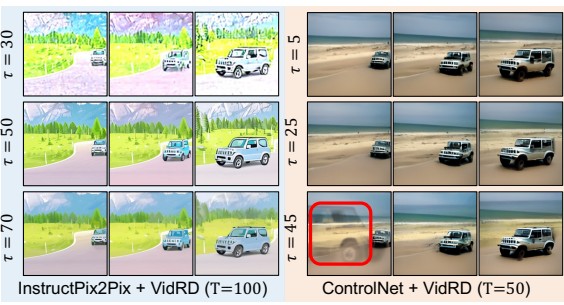

Figure 9: Effects of $\tau$, the timestep that FLDM starts to fuse T2V latents with T2I latents during the denoising process.

Table 3: Ablation study of different $\tau$.

| Method | timestep($\tau$) | CLIP Mertics Tem-Con | CLIP Mertics Text-Align | PickScore User-Pre |
|---|---|---|---|---|
| VidRD + IP2P | 30 | 88.73 | 24.89 | 20.42 |
| | 50 | **89.09** | **24.95** | **20.49** |
| | 70 | 88.11 | 24.79 | 20.43 |
| VidRD + CN | 5 | 86.59 | **27.55** | 20.60 |
| | 25 | **90.14** | 27.43 | **20.64** |
| | 45 | 82.60 | 25.41 | 20.14 |

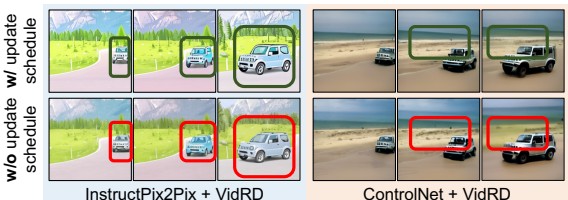

Figure 10: Effectiveness of update schedule for fusion ratio $\alpha$, examples show it can improve temporal consistency while reducing image artifacts.

appearance of the car (Left: InstructPix2Pix + VidRD) and the beach (Right: ControlNet + VidRD) can be better preserved among video frames with the update schedule. This illustrates that, with decayed T2I latent proportion at the last $T - \tau$ timesteps, the update schedule further reduces T2I latents' negative effect on temporal consistency.

## 5 CONCLUSION

In this paper, we propose FLDM (Fused Latent Diffusion Model), a simple-yet-effective strategy that achieves high video editing quality without any tuning cost. For each denoising timestep, we apply a hyper-parameter to adjust the latent ratio of T2V and T2I diffusion models, with an update schedule to alleviate image artifacts. For the first time, we reveal that T2V and T2I diffusion models are complementary to each other in temporal consistency and structure. Our method can serve as a versatile plugin for various off-the-shelf T2I and T2V models, which we believe will be valuable for real-world practice.

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
