# OpenReview forum: "Fuse Your Latents: Video Editing with Multi-source Latent Diffusion Models"
_acmmm.org/ACMMM/2024/Conference — MM2024 Poster_

### Official Review · Reviewer_a5th · 2024-05-04

**Rating:** 4
**Confidence:** 3

**Summary:**

This paper proposes a train-free framework on fusing the T2V and T2I LDMs to edit the video. They use weighted sum to fuse these two. This fusion method tries to make a balance between structural modeling and temporal consistency. An update schedule on the weight $\alpha_t$ are also proposed to firstly get a good structure and then obtain temporal consistency. The results can show the trade-off between T2I and T2V models and show the effectiveness of the proposed methods.

**Strengths:**

1.It’s a good exploration about cooperation between T2I and T2V LDMs, revealing the strength and weakness of the two kinds.
2.To a certain extent, the proposed method can integrate the advantages of T2I and T2V LDMs, without any training period.
3.A reasonable update schedule is designed to get the structural modeling firstly and then obtain the temporal modeling, which makes sense.

**Limitations:**

1.According to the results shown in Tab. 1, performance of the proposed method is not very satisfactory for me. Compared with VidRD & DDIM and ZS & DDIM, the proposed method has a big drop (about 2 points) on CLIP Tem-Con and a negligible improvement on CLIP Text-Align.
2.Lack the comparison with some SOTA video editing method, like Video-P2P, ControlVideo.
3.As for the qualitative results shown in Fig.6 left, the structural modeling of ZS+IP2P is not good (the “dog” is still similar to a rabit).
4.Lack the quantitative ablation study on the update schedule, I think it important to show this for comprehensive validity provement.

**Suitability:**

3

---

### Official Review · Reviewer_9BZw · 2024-05-16

**Rating:** 4
**Confidence:** 4

**Summary:**

This paper proposes to tackle the problem of video editing via integrating various T2I and T2V LDMs, named FLDM. Specifically, FLDM utilizes a hyper-parameter with an update schedule to effectively fuse image and video latents during the denoising process. Experimental results show effectiveness of proposed method.

**Strengths:**

1. The writing is clear, logical, and easy to understand.
2. The proposed method is simple yet efficient.
3. Experimental results are complete and comprehensive.

**Limitations:**

1. The proposed method is somehow tricky. Since the chosen T2I model is ControlNet and InstructPix2Pix as written in L515, it's pretty much like the ControlVideo [5] that can already preserve the structure information. The chosen video model is a pre-trained pure text-to-video model without any constraints of structure information. If we just use a structure preserve T2V model such as [3,5], then the hypothesis of T2V's non-structure preserving property is no longer valid.
2. About experimental results, lacking of comparison with other SOTA video editing methods [1-3].

Additional Information:
1. Is there any reason or experiment to explain that T2V cannot preserve structure?


[1] Geyer M, Bar-Tal O, Bagon S, et al. Tokenflow: Consistent diffusion features for consistent video editing[J]. arXiv preprint arXiv:2307.10373, 2023.
[2] Kara O, Kurtkaya B, Yesiltepe H, et al. RAVE: Randomized Noise Shuffling for Fast and Consistent Video Editing with Diffusion Models[J]. arXiv preprint arXiv:2312.04524, 2023.
[3] Feng R, Weng W, Wang Y, et al. Ccedit: Creative and controllable video editing via diffusion models[J]. arXiv preprint arXiv:2309.16496, 2023.
[4] Zhang Y, Wei Y, Jiang D, et al. Controlvideo: Training-free controllable text-to-video generation[J]. arXiv preprint arXiv:2305.13077, 2023.
[5] Liew J H, Yan H, Zhang J, et al. Magicedit: High-fidelity and temporally coherent video editing[J]. arXiv preprint arXiv:2308.14749, 2023.

**Suitability:**

3

---

### Official Review · Reviewer_QYw9 · 2024-05-23

**Rating:** 3
**Confidence:** 4

**Summary:**

This work achieves high-quality T2V editing by integrating various T2I and T2V LDMs. Specifically, FLDM utilizes a hyper-parameter with an update schedule to effectively fuse image and video latents during the denoising process.

**Strengths:**

1. This paper is the first to reveal that T2I and T2V LDMs can complement each other in terms of structure and temporal consistency, ultimately generating high-quality videos.

2. FLDM can serve as a versatile plugin, applicable to off-the-shelf image and video LDMs, to significantly enhance the quality of video editing.

**Limitations:**

1. It is a natural idea to introduce the video diffusion model as the basic model into video editing. However, this article seriously ignores the focus of this type of improvement, that is, 1) it does not provide a detailed description of what motivation it is based on, 2) why it is introduced in this way, and 3) design multiple introduction methods and explore their differences.

2. In addition, the comparison method was too early, and there was even no comparison of papers in NeurIPS2023.

**Suitability:**

3

---

### Official Review · Reviewer_BdtJ · 2024-05-25

**Rating:** 3
**Confidence:** 3

**Summary:**

This paper proposes the Fused Latent Diffusion Model, a video editing technique that uses T2I and T2V models together for better temporal consistency and structure fidelity. The latent fusion schedule is carefully designed to balance the influence of each independent component. The results show that the proposed method helps overcome the limitations of the baselines.

**Strengths:**

1. The supplementary project page provides videos that help understand the method;
2. The method is technically clear and easy to follow;
3. The paper is well-constructed and easy to understand.

**Limitations:**

1. The qualitative result is limited. The supplementary material only provides less than ten edited videos, making it hard to tell the generalizability of the proposed method;
2. According to all samples, the editing capability seems limited. Most results (especially the "car driving on the road" example) share almost the same structure as the input videos and only perform texture-level modification. Is the proposed method able to do some sematic-level editing, e.g., changing the car into a Lamborghini?
3. In Tab. 1, why FateZero achieve the highest temporal consistency and outperform the proposed method?
4. The compared baselines are not comprehensive. Should additionally consider other representative works such as TokenFlow, Pix2Vid, and CoDeF.

**Suitability:**

2

---

### Meta-Review · Area_Chair_pQTP · 2024-07-01

**Recommendation:** Accept (Poster)
**Confidence:** 4

**Metareview:**

This work proposes a fused latent diffusion model that combines T2I and T2V pretrained models for video editing. After the rebuttal, mouse reviewers are positive about this work. It is worth mentioning that the proposed method is simple yet effective, with comprehensive experimental results. Thus the AC recommends the acceptance of this paper. It is highly recommended to include more qualitative visual results in the revised version of the paper.